

# Large scale climate response of the Southern Ocean and Antarctica to reduced ice sheets

Katherine Power[1], Fernanda Matos[2], and Qiong Zhang[1]

[1]Department of Physical Geography and Bolin Centre for Climate Research, Stockholm University, Stockholm, Sweden
[2]Alfred Wegener Institute - Helmholtz Centre for Polar and Marine Research, Bremerhaven, 27570, Germany

**Correspondence:** Katherine Power (katherine.power@natgeo.su.se)

**Abstract.** The Antarctic and Greenland Ice Sheets (AIS and GIS) are critical tipping points in the Earth's climate system. Their potential collapse could trigger cascading effects, significantly alter the global climate patterns, and cause large-scale, long-lasting, and potentially irreversible changes within human timescales. This study investigates the large-scale climate response of the Southern Ocean and Antarctica to the isolated effect of reducing ice sheet extent. Using well-documented
paleogeography of the Late Pliocene (LP) from the PRISM4D reconstruction, where the West Antarctic Ice Sheet (WAIS) and the northwestern GIS were significantly diminished, we conducted 1450-year simulations with the EC-Earth3 model at atmospheric $CO_2$ concentrations of 280 ppmv and 400 ppmv.

Our results reveal that the implementation of the LP ice sheet configuration leads to a 9.5°C rise in surface air temperature, approximately 16% reduction in sea ice concentration, and a 0.63 mm/day increase in precipitation over Antarctica and the
Southern Ocean. These changes far exceed those driven by $CO_2$ increase alone, which result in a 2.5°C warming and a 9.3% sea ice decline. Throughout the simulations, the positive phase of the Southern Annular Mode (SAM) persists, intensifying westerly winds and contributing to sea ice export and deep ocean warming. The combined effects of ice sheet reduction and $CO_2$ forcing initially weakened Antarctic Bottom Water formation, with major implications for the Global Meridional Overturning Circulation due to stronger water column stratification driven by surface freshening and subsurface warming.

By isolating the direct albedo effect of reducing GIS and AIS extents, this study offers critical insights into the mechanisms driving atmospheric and oceanic variability around Antarctica and their broader implications for global climate dynamics. Although the complete Late Pliocene boundary conditions may serve as a valuable analogue for current and future climate change, excluding the orographic changes allows us to specifically assess the primary role of surface reflectivity. This targeted approach lays the groundwork for future research to explore how the combined effects of albedo and paleogeographical changes
could influence interhemispheric climate feedbacks under scenarios of future ice sheet collapse or instability.

## 1 Introduction

The Greenland and Antarctic ice sheets (GIS and AIS hereafter) are pivotal components of the Earth's climate system. Their high albedo reflects a significant portion of solar radiation, thereby cooling surrounding regions and playing a critical role in regulating global air and sea surface temperatures. In addition, these ice sheets influence atmospheric and oceanic processes by





driving sea ice and dense water formation, which are essential elements of global thermohaline circulation (Clark et al., 1999). However, the stability of these ice sheets is increasingly at risk.

Projections indicate that the accelerated melt of the GIS and AIS, coupled with the retreat of their associated ice shelves (Naughten et al., 2023; Greene et al., 2024), will have far-reaching implications for global climate dynamics (Buizert et al., 2018). These changes are likely to disrupt critical processes such as deep-water formation and the contribution of the southern

ocean to global heat and carbon transport. Understanding these risks is critical, as the potential collapse of the AIS and GIS could trigger cascading climate feedbacks, leading to irreversible and long-lasting changes within human timescales.

Paleoclimate records offer unique insights into the behavior of these ice sheets during past warm periods, such as Marine Isotope Stage 5e (~125,000 years ago) in the Pleistocene (Steig et al., 2015) and the Late Pliocene epoch (LP ~3,3 million years ago) (Naish et al., 2009; Kim and Crowley, 2000). These periods were characterised by significantly smaller ice sheets than today. Using these analogs, we can better understand how ice sheet dynamics influence oceanic and atmospheric processes during these past warm periods, which can offer valuable lessons for predicting future climate behavior as the Earth continues to warm.

In this study, we apply Late Pliocene ice sheet configurations to a modern geographic framework using sensitivity experiments with the EC-Earth model. This approach allows us to assess the sensitivity of the climate system to changes in ice

sheet extent and varying $CO_2$ concentrations, using ice sheet conditions of the past as an analogue for the future. By focusing on the albedo effect and excluding orographic changes, our goal is to uncover the key mechanisms and processes that could profoundly influence Earth's future climate, environment, and societies.

## 2  Model configuration and experiments setup

### 2.1  Model configuration

We use the low-resolution configuration of the EC-Earth model, EC-Earth3-LR, an Earth System Model (ESM) developed collaboratively by the European research consortium EC-Earth. EC-Earth model has flexible configurations that allow for the inclusion or exclusion of various climate processes, making it a versatile tool for a wide range of climate studies (Döscher et al., 2022). EC-Earth3 integrates several key components, including the atmospheric model IFS cycle 36r4, the land surface module HTESSEL, the ocean model NEMO3.6 and the sea-ice module LIM3, all coupled via the OASIS3-MCT coupler. IFS

and HTESSEL have a horizontal linear resolution of TL159 (1.125°), and the ocean and sea-ice components (NEMO and LIM) have a nominal resolution of 1° (Döscher et al., 2022).

The low-resolution configuration was selected to significantly reduce computational demands, allowing for conducting multi-century simulations and various sensitivity experiments. This setup is particularly suited for exploring slow processes in the deep ocean, which are central to the goals of this study. Such processes include changes in stratification, overturning

circulation, and the response of Antarctic Bottom Water formation to altered climate forcing.

The EC-Earth3 model has consistently demonstrated its effectiveness in capturing key climate dynamics, including temperature variability, heat fluxes and other essential aspects of the Earth's System. This capability facilitates a more comprehen-



sive understanding of the impacts of natural and anthropogenic forcing on the global climate system (Koenigk et al., 2013; Döscher et al., 2022; Cao et al., 2023). EC-Earth3 has been extensively validated in both modern and paleo-climate studies,

showing robust performance in simulating the climates of past warm periods such as mid-Holocene, Last Interglacial and mid-Pliocene epochs. These simulations have provided valuable information that has been integrated to major model inter-comparison projects, such as PMIP4 (Paleoclimate Model Intercomparison Project phase 4) and PlioMIP2 (Pliocene Model Intercomparison Project phase 2) (Zhang et al., 2021; Chen et al., 2022; Power et al., 2023; Han et al., 2024)

## 2.2   Experiments setup

To investigate the impacts of varying ice sheet configurations and $CO_2$ concentrations in the Southern Ocean and Antarctica, we performed a series of sensitivity experiments. These experiments employed modern ice-sheet configurations (labeled E) and Late Pliocene ice-sheet reconstructions (labeled Ei) under two atmospheric $CO_2$ levels: pre-industrial (280 ppmv) and intermediate (400 ppmv).

   The Late Pliocene Antarctic ice sheet reconstruction used in these experiments (Haywood et al., 2016; Chandan and Peltier,

2018) were originally developed using the high-resolution British Antarctic Survey Ice Sheet Model, integrated with clima-tologies from the Hadley Centre Global Climate Model (Hill et al., 2007; Hill, 2009), utilising PRISM2 boundary conditions (Dowsett et al., 1999). The mid-Pliocene Greenland Ice Sheet (GIS) reconstruction provided for PlioMIP2 Haywood et al. (2016) is based on 30 modelling results from the PLISMIP project Dolan et al. (2012). Given the focus of this study on the domain south of 60ºS (Antarctic sector hereafter), the effects of Greenland ice sheet are not explored in depth.

To isolate the impact of surface reflectivity (albedo), we excluded orographic changes and freshwater input associated with reduced ice sheet extent. This decision enables a targeted investigation of energy balance changes solely by surface albedo. Figure 1 provides a visual comparison of the modern and Pliocene ice sheet configurations.

   These simulations are part of the Pliocene for Future (P4F) Tier 2 experiments (Haywood et al., 2016), which has been proposed for the second phase of the Pliocene Model Intercomparison Project (PlioMIP2) and will be included in the new

PlioMIP3 experiment list. The reliability of the Antarctic ice sheet configuration is further supported by the results of the PLISMIP results, which evaluated the dependencies of the ice sheet model for the warm period of the mid-Pliocene using 30 different models (Dolan et al., 2012). To ensure consistency across all simulations, modern vegetation, as simulated for the year 1850 CE, was fixed using the off-line LPJ-GUESS dynamic vegetation model (Chen et al., 2021). Each simulation spans a minimum of 1450 years, with the final 200 years of model output used for analysis of the mean state. The pre-industrial

configuration (E280) serves as the PI control experiment for comparison.





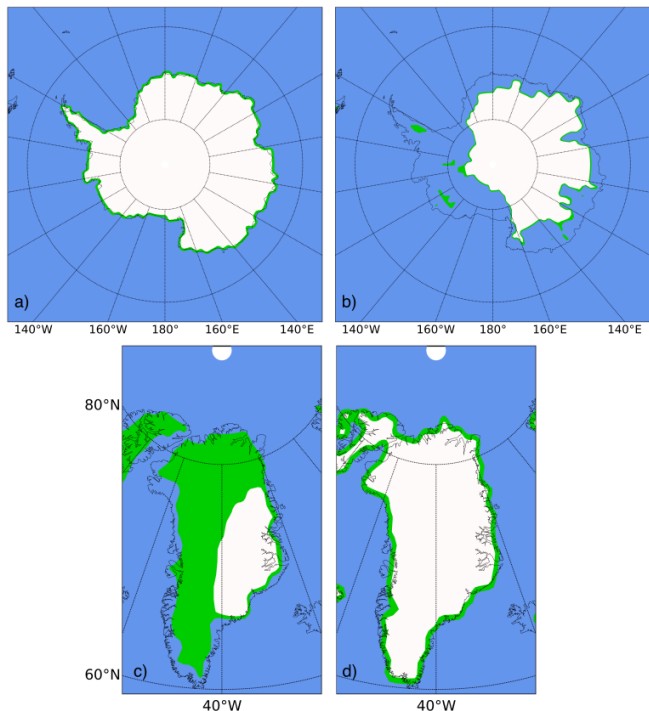

**Figure 1.** Comparison of the modern and Pliocene ice sheets as provided by the PLISMIP project.

## 3 Antarctica and Southern Ocean response to $CO_2$ forcing and ice sheet reduction

### 3.1 Near Surface Processes

The interactions between the atmosphere, cryosphere and ocean are crucial in understanding how changes in $CO_2$ levels and ice sheet configurations influence climate feedbacks. Surface warming driven by both atmospheric and albedo changes directly impacts oceanic processes, particularly in the Southern Ocean. The delicate balance between surface stratification and deep water formation in this region is critical to regulating global ocean circulation.

The simulations reveal that the near surface air temperature (TAS) increases by 2.51°C under increased $CO_2$ (E400 relative to E280; Figure 2a). However, when Pliocene ice sheet configurations are applied (Ei400 relative to E280), the warming intensifies, reading 9.49°C (Figure 2b). Similarly, sea surface temperature (SST) increased by 1.26°C under $CO_2$ forcing alone but increased to 4.89°C with the combined effects of $CO_2$ and ice sheet changes (Figure 3a and b), compared to PI. These results underscore the amplified warming effect of reduced ice sheet extent compared to $CO_2$ increase alone.

The spatial pattern of TAS and SST warming strongly align with albedo changes ( Figures2a and b, and 3a and b). Under elevated $CO_2$ (E400), albedo reductions are only due to sea ice loss, primarily in the Ross and Weddell Seas, leading to a



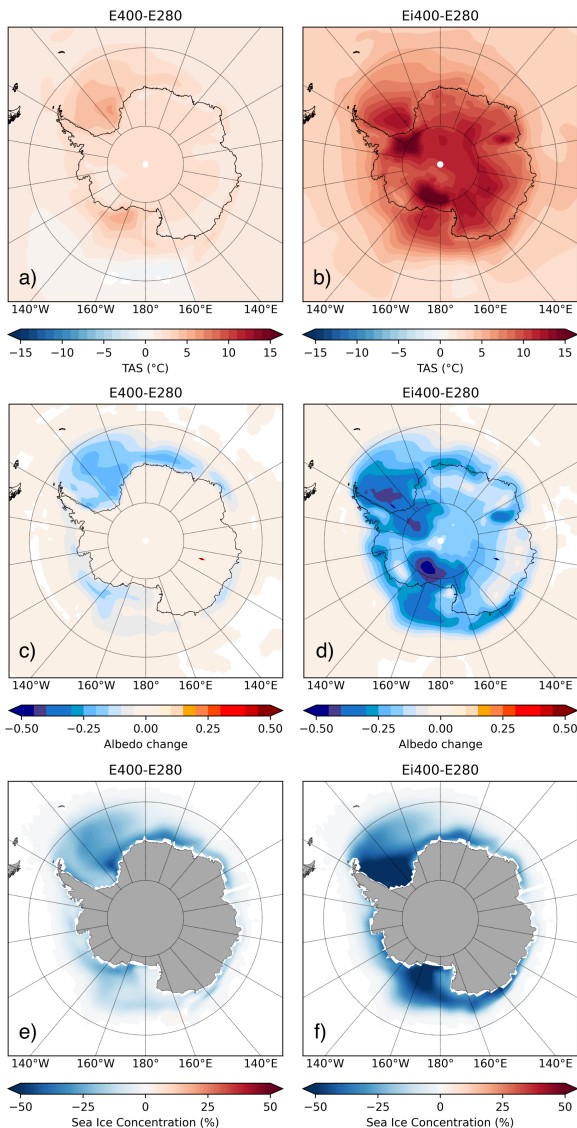

**Figure 2.** Temperature, sea ice concentration and albedo anomalies resulting from increased $CO_2$ levels and the reduction of the modern ice sheet to Pliocene size. a) E400-E280 surface air temperature, b) Ei400-E280 surface air temperature, c) E400-E280 albedo, d) Ei400-E280 albedo, e) E400-E280 sea ice concentration, f) Ei400-E280 sea ice concentration. White areas indicate results are not statistically significant at the 95% confidence level.

9.25% decline in sea ice concentration (Figure 2e). In contrast, the configuration of the Pliocene ice sheet induces a widespread decline in albedo throughout the Antarctic, resulting in a more pronounced loss of sea ice of 16. 25% (Figure 2 f). The loss of sea ice further reinforces the warming through a positive albedo feedback mechanism, exposing more of the ocean surface to direct solar heating.



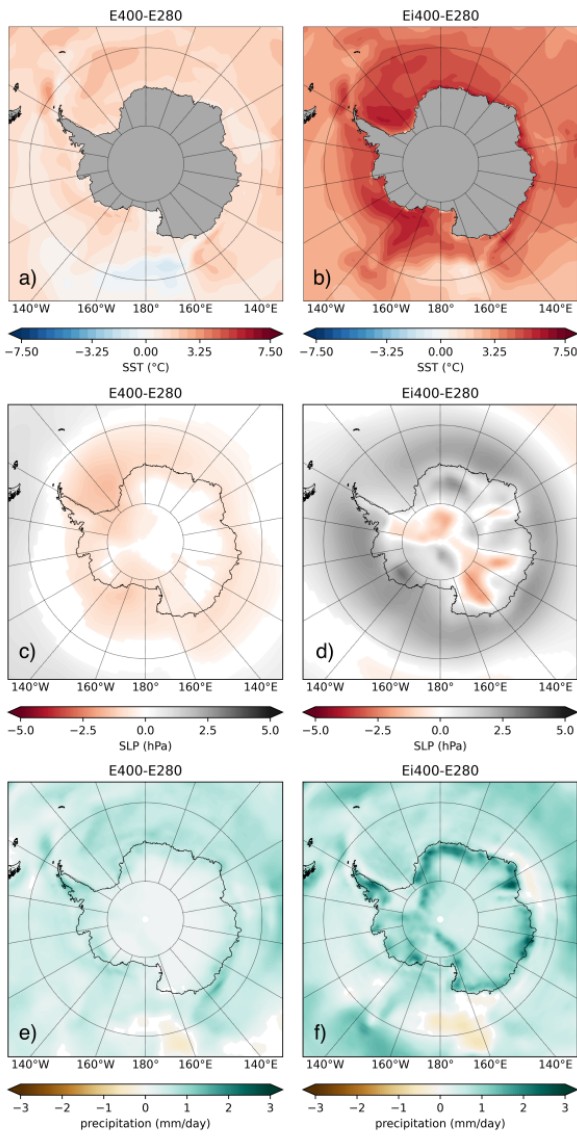

**Figure 3.** Sea Surface Temperature (SST), sea level pressure (SLP) and precipitation anomalies resulting from increased CO$_2$ levels and the reduction of the modern ice sheet to Pliocene size. a) E400-E280 SST, b) Ei400-E280 SST, c) E400-E280 SLP, d) Ei400-E280 SLP, e) E400-E280 precipitation, f) Ei400-E280 precipitation. White areas indicate results are not statistically significant at the 95% confidence level.

Precipitation patterns also exhibit significant changes. Under increased CO$_2$, precipitation increases by 0.42 mm/day relative to the PI control simulation, with the highest increases along Antarctic coasts (Figure 3e). When ice sheet reductions are included, precipitation increases further to 0.63 mm/day (Figure 3f), driven by enhanced atmospheric moisture transport. These






changes are closely related to a persistent positive phase of the Southern Annular Mode (SAM) (Figure 4), which intensifies the westerly winds and enhances moisture convergence around the Antarctic.

The positive SAM phase also influences the Sea Level Pressure (SLP) patterns. Elevated $CO_2$ levels lead to a mean reduction in SLP of 0.55 hPa over Antarctica and the surrounding Southern Ocean, reflecting the weakened high-pressure system due to

atmospheric warming (Figure 3c). Conversely, reduced ice sheet extent results in a mean SLP increases of 1.37 hPa in most of Antarctica Figure 3d), strengthening the pressure gradient between the pole and mid-latitudes and intensifying westerly winds.

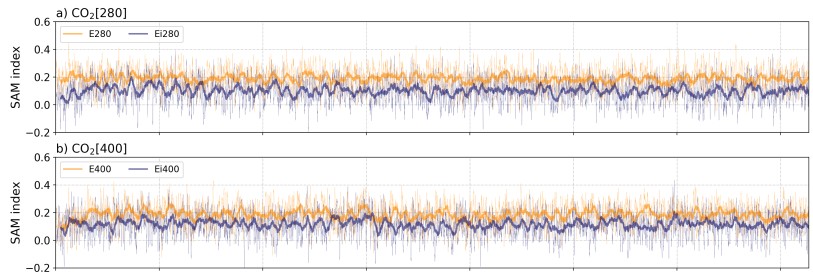

**Figure 4.** Southern Annular Mode (SAM) index timeseries for the E and Ei experiments. a) At 280 ppmv $CO_2$. b) At 400 ppmv $CO_2$). Thinner lines represent annual mean, while thicker lines are applied 10-year running to show the decadal variability

This phase indicates a contraction of westerly winds towards Antarctica, causing an eastward shift and a deepening of the Amundsen sea low (Goddard et al., 2021), with stronger winds and stormier conditions in the Southern Ocean, the Antarctic Peninsula, the Bellingshausen Sea and the eastern Amundsen Sea (Fogt et al., 2011; Hosking et al., 2013; Raphael et al., 2016).

Consequently, the influx of warm and moist maritime air into the west Antarctic increases precipitation while also contributing to regional warming and sea ice loss. The deepening of the Amundsen Sea Low enhances the northward export of sea ice in the Ross Sea. Simultaneously, the poleward contraction of westerlies drives increased upwelling of warm, deep ocean waters in the Ross Gyre region. This upwelling accelerates sea ice melting from below, further amplifying surface warming due to sea ice loss, as seen in Figure 2. The loss of sea ice further lowers surface albedo, creating a positive feedback loop that

accelerates warming. The exposed ocean surface absorbs more solar radiation, further intensifying the sea ice-albedo feedback and amplifying regional warming.

### 3.2 Southern Ocean sensitivity to reduced AIS

Reducing the extent of the AIS has significant implications for Southern Ocean processes, particularly sea ice dynamics and deep-water formation. Although our model does not account for marine ice sheet instabilities, the loss of surface reflectivity

and resulting albedo feedbacks are sufficient to induce notable changes in the Southern Ocean's stratification and overturning circulation.

At the surface, the reduction in sea ice is apparent. In the Ei400 experiment, the concentration of sea ice decreases by greater than 25% compared to PI (Figure 2 f), for some regions, exposing more of the ocean surface to direct solar heating. This warming contributes to a positive feedback loop, where the loss of sea ice lowers albedo, accelerates surface warming, and





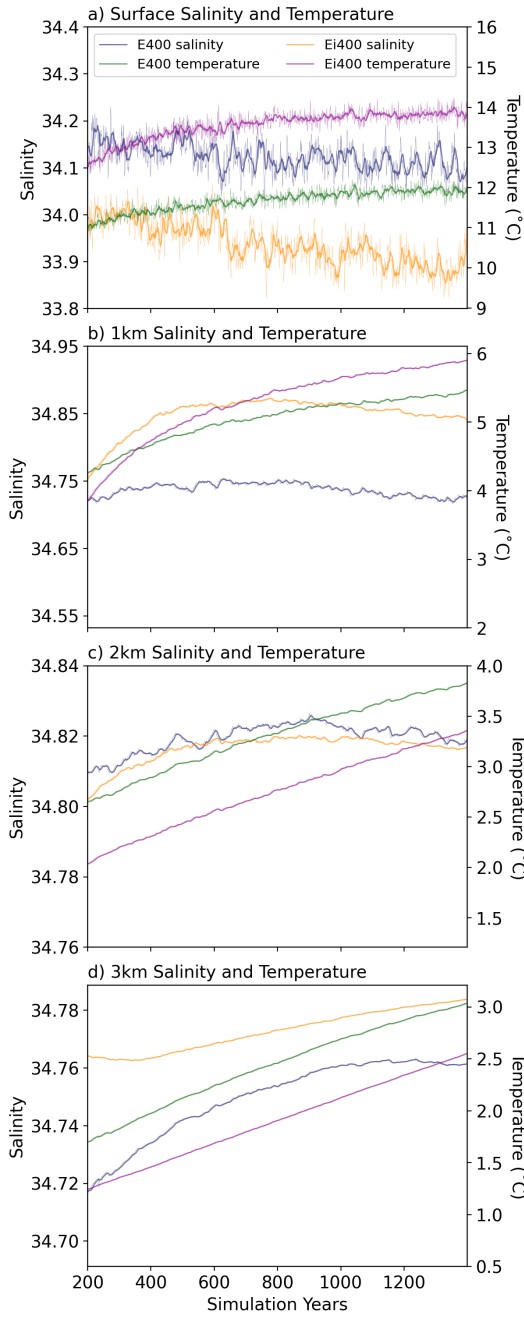

**Figure 5.** Time series of the temperature and salinity of the Southern Ocean at a) surface, b) 1km, c) 2km and d) 3km depths.

further reduces the extent of the sea ice. The sea surface temperature (SST) in Ei400 increases by up to $1^oC$ relative to the start of the experiments. The salinity and temperature time series in figure 5 exhibit interannual to decadal variability, which may be



linked to the interdecadal Pacific oscillation that dominates the variability of Southern Ocean SSTs on such a timescale (Yao et al., 2024).

Freshening of the Southern Ocean is another prominent feature of the reduced AIS experiment. The combination of enhanced
precipitation and sea ice melt dilutes surface salinity, creating a stratified upper ocean layer that inhibits vertical mixing and deep water formation (Figure 5 a). This stratification has profound effects on deep convection processes, which are essential to ventilate the southern ocean and maintain the strength of Antarctic Bottom Water (AABW) formation.

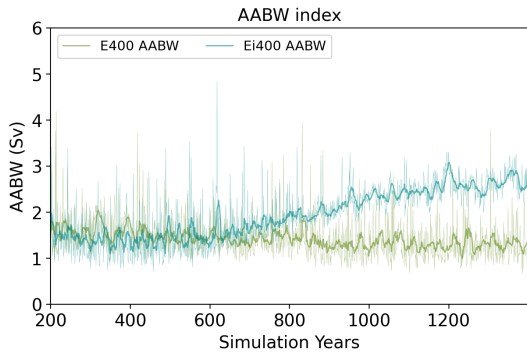

**Figure 6.** Time series of the Antarctic Bottom Water formation index for experiments E400 and Ei400, calculated as the absolute value of the minimum global streamfunction south of 60$^o$S and below 500 m depth (adapted from Zhang et al. (2019)).

Descending through the water column, temperature and salinity changes exhibit distinct patterns across depths. At 1 km, 2 km and 3 km depths, the seawater temperature increase by up to $1.5^o$ relative to the beginning of the simulation (Figure
5. While surface layers show substantial freshening, deeper layers undergo initial salinisation for the first 700 years of the simulation, followed by a transition to freshening. This transition reflects the dynamic interaction between surface buoyancy fluxes, vertical mixing, and the redistribution of heat and salt within the Southern Ocean. Since neither the wind regime nor the Southern Ocean currents shifted significantly during our simulations, the thermohaline changes observed are primarily driven by the sea-ice-albedo feedback and the atmospheric greenhouse gas forcing.

The influence of the Antarctic continent on global ocean circulation is largely mediated through its intermediate and bottom water masses. Antarctic Intermediate Water (AAIW), and Antarctic Bottom Water (AABW). AABW, in particular, plays a critical role as it ventilates all major ocean basins Orsi et al. (1999). Understanding the processes governing its formation and variability is therefore essential for assessing the broader impacts of climate change.

A key consequence of these processes is the suppression of AABW formation. In both the E400 and Ei400 experiments,
AABW strength, reduced by approximately 1 Sv ($10^6 m^2 s^{-1}$) by year 700 of the simulation compared to PI control (2 Sv) (figure 6). This reduction signifies a weakening of deep-water formation caused by increased stratification and reduced brine rejection during sea ice formation.

Interestingly, the AABW trends diverge in the later stages of the simulation. In the E400 experiment, AABW continues to weaken, whereas in Ei400 experiment, it begins to recover after year 700. This recovery suggests that the reduced AIS triggers



a compensatory mechanism, likely involving the import of salinity into the Southern Ocean, which enhances AABW strength
       on multi-centennial timescales, counteracting the initial suppression.

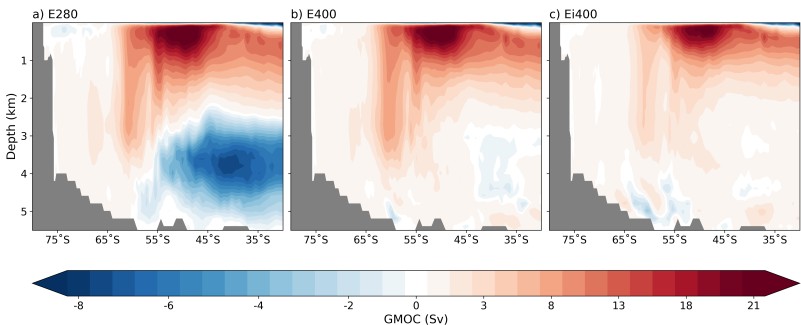

**Figure 7.** Southern Ocean MOC for the a) E280, b) E400, and c) Ei400 experiments during averaged over the last 200 years of the simulation

       The broader impact for Global Meridional Overturning Circulation (GMOC) is evident in figure 7. The AABW cell, repre-
       sented by the blue anticlockwise circulation at the bottom of the ocean, shrinks significantly in E400 but retains greater strength
       and coverage in the Ei400 experiment (figures 7b and 7c). This pattern aligns with the findings of (Sidorenko et al., 2021),
where surface buoyancy loss, ozone depletion, and stronger westerlies over the Southern Ocean inhibit AABW formation and
       export. However, our simulations suggest that the reduced AIS not only amplifies surface-driven feedbacks but also initiates
       deep-ocean processes that partially mitigate the suppression of deep-water formation.

       In particular, our simulations do not indicate significant changes in the wind regime over Antarctica or the Southern Ocean.
       This absence suggests that the observed effects are mainly driven by changes in surface buoyancy due to the combined impact
of $CO^2$ forcing and ice sheet loss. In contrast to scenarios where strengthened winds amplify surface buoyancy loss, our model
       reproduces a similar state through thermohaline feedbacks alone. Furthermore, removing such a large extent of the AIS induces
       an amplified buoyancy gain response in the deep ocean, which counterbalances the strengthening of the upper overturning cell
       over multi-centennial timescales. This response highlights the role of salinity import in enhancing the strength of AABW under
       the condition of reduced ice sheet extent.

In general, ocean warming driven by a negative sea-ice albedo feedback increases stratification at deep convection sites,
       inhibiting the vertical mixing required for deep-water formation. This self-reinforced mechanism amplifies the positive phase
       of the SAM, intensifying the effects of atmospheric circulation that contribute to regional warming and increased precipitation
       around Antarctica at elevated $CO_2$ levels. However, these processes are insufficient to fully balance the salinisation that occurs
       in the deep ocean, which remains a key challenge for the recovery of deep-water formation.

Although our simulations represent idealised scenarios, the results underscore the critical sensitivity of the Southern Ocean to
       increased atmospheric warming. Current conditions do not yet reflect the full extent of AIS reduction or collapse, as projected
       in future climate experiments Roach et al. (2023); Armstrong McKay et al. (2022); Naughten et al. (2023); Steig et al. (2015).
       However, our results suggest that even under present trends, the ability of the Southern Ocean to ventilate the deep ocean is
       at significant risk. Furthermore, while we isolate the albedo effect in this study to reduce uncertainties, the exclusion of other



climate feedbacks may underestimate the potential catastrophic outcomes of AIS collapse. These insights are vital for tuning climate models and reconciling data-model discrepancies.

## 4 Late Pliocene ice sheets as analogues for future climate

The GIS and AIS are undergoing dramatic changes due to polar amplification (Armstrong McKay et al., 2022). If the current pace of radiative forcing continues, the potential collapse of these ice sheet could trigger complex climate feedbacks. These
include increased meltwater input, surface cooling, changes in westerly wind patterns, and multi-centennial variability in sea-ice production. Although some feedbacks may eventually promote the recovery of deep-water production when a certain threshold is reached (Johnson et al., 2024; Aylmer et al., 2022; Kang et al., 2023), the pathways to such recovery remain highly uncertain.

There is a notable gap in existing research on the isolated impact of the abrupt removal of large portions of AIS and GIS on
climate and ocean circulation. The Late Pliocene epoch serves as an important analogue of future climate scenarios due to its modern-like atmospheric $CO_2$ concentrations, significantly reduced ice sheets, comparable ocean gateway configurations, and ecosystem shifts. This paleogeographic framework offers a valuable opportunity to assess climate sensitivity to regional albedo changes. In this study, we specifically isolate the albedo effect of ice sheet reduction to examine its role in driving Antarctic climate dynamics and Southern Ocean circulation. Our simulations with PRISM4D ice sheet conditions demonstrate that ice
sheets play a critical role in modulating climate feedbacks in response to warming.

The configuration of the Pliocene ice sheet results in a substantial Antarctic warming of 9.5°C, an increase in SST by 4.9°C, a loss of 16. 2% sea ice, and a rise in precipitation by 0.63 mm/day. As expected, the atmospheric and oceanic responses observed in our sensitivity experiments do not fully reproduce the climate changes seen in more comprehensive modeling studies that incorporate all boundary conditions of the Late Pliocene. Nor do they match the reconstructed climate based on
proxy data (Burls et al., 2017; Haywood et al., 2013, 2016, 2024). However, current and future climates are not exact replicas of the Late Pliocene either. Thus, our conclusions focus on the idealised interactions of climate feedbacks in these controlled experiments.

Our decision not to include the orographical change associated with reduced ice sheets is based on two considerations: (1) There is no clear indication that future orographical changes will closely align with those reconstructed for the Late Pliocene,
even though strong evidence suggests ice sheet extent may be similar; (2) Modifying orography in models is a complex task that involves adjusting not only to mean orography in the atmospheric component but also sub-grid scale parameters, such as standard deviation, slopes, and angles. Deriving these parameters introduces additional uncertainties that could outweigh the benefits of including orographic changes in an idealised sensitivity experiment.

Thus, we conclude that the reduction of AIS primarily influences the circulation of the Antarctic and Southern basins. By
isolating the albedo effect, our study provides a foundational understanding of how ice sheet loss, independent of freshwater input and orographic changes, can significantly alter Southern Hemisphere climate dynamics. These insights are critical for



refining future climate models and identifying early signals of ice sheet retreat, offering a clearer picture of the potential pathways and risks associated with polar ice sheet instability.

*Author contributions.* Conceptualization, K.P., F.M., Q.Z.; methodology, K.P., F.M.; formal analysis, K.P., F.M.; investigation, K.P., F.M.;

resources, K.P., Q.Z.; data curation, K.P.; writing–original draft preparation, K.P.; writing–review and editing, K.P., F.M., Q.Z.; visualization, K.P. F.M.; project administration, Q.Z.

*Competing interests.* The authors declare no conflict of interest.

*Acknowledgements.* This work was supported by the Swedish Research Council (Vetenskapsrådet, grant no. 2022-03129)

The data analyses were performed using resources provided by the ECMWF's computing and archive facilities and Swedish National

Infrastructure for Computing (SNIC) at the National Supercomputer Centre (NSC), which is partially funded by the Swedish Research Council through grant agreement no. 2022-06725.



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
