# Peer review of "Late Pliocene Ice Sheets as an Analogue for Future Climate: A Sensitivity Study of the Polar Southern Hemisphere"

_EGUsphere, 2024_

## Author Response (AR1)

**Author response for egusphere-2024-4061**

Katherine Power, Fernanda Matos, Qiong Zhang

April 21, 2025

Dear Editors and Reviewers,

We are extremely grateful to the editor for collating the reviews and the reviewers for thorough evaluations and constructive feedback. We have taken on board every one of their concerns and used them to, hopefully, improve the manuscript to publication level. The lengthy and in depth comments have been invaluable in guiding us as to how to properly address this important subject, going deeper into analysis and exploring more variables to provide a more coherent and investigative picture.

We have undertaken a large rewritting of the paper. In its new form we have included:

- Introduction is now improved with a concise literature review.
- A coherent description of the aims of our study which recognises the importance and necessity of this
  work.
- Methodology comprehensive experiment outlines detailing the full procedure undertaken to run these sensitivity experiments.
- Results section now includes proper SAM analysis and investigation in the westerly jet. We provide
  evidence for all claims.
- Thorough discussion section linking the processes we found, supported by appropriate references.
- Pliocene as analogue section now compares our results with relevant work, plus concise limitations section.
- Conclusion section sets out future work that would add to this study.

We hope we have truly addressed all the concerns and produced a more complex paper that advances our knowledge of this important subject matter. Below follows a reply to each reviewer comment.

> Sincerely, Authors

**Reviewer 1**

1. It should be made clearer in the title and first part of the abstract that only albedo changes and not orographic changes are taken into account.

**Response#: We have revised the manuscript to ensure the emphasis is in isolating the surface reflectivity (albedo) effect of reduced ice sheet extent, excluding orographic and freshwater feedbacks. This includes changing the title to "Late Pliocene Ice Sheets as an Analogue for Future Climate: A Sensitivity Study of the Polar Southern Hemisphere" and modifying the abstract and other sections of the manuscript.**

2. Improved methods: It might be helpful to include a table of the experiments performed with the forcing included. I am actually surprised to see changes in albedo over the East Antarctic ice-sheet in fig. 2d. It is my understanding the ice-sheet is a forcing in this simulation, and thus the albedo over the ice-sheet imposed. Can the authors explain why albedo is changing over land areas where the ice-sheet forcing should not have changed?

**Response#: A table has now been included to section 2.2 that gives full details of the experiments and the forcings. We also give a full description of the pre-industrial experiment design:**

"The protocol for our pre-industrial (PI) simulation follows Eyring et al. (2016). Ice sheets, land geography, topography and vegetation are all unmodified from the model. GHG concentrations for  $CO_2$ ,  $CH_4$  and  $N_2O$  are 284.3 ppmv, 808.2 ppbv, and 273.0 ppbv, respectively. For orbital parameters; eccentricity set at 0.016764, obliquity 23.549 and perihelion - 180 is 100.33."

Table 1: The four Core and Tier 2 Pliocene for Future protocol experiments conducted. PI refers to preindustrial conditions, LP for Late Pliocene. Name terminology is from Haywood et al. (2016).

| Experiment ID | Ice sheet Configuration | Land Geography | Topography | Vegetation | $CO_2$ (ppm) | Orbit |
|---------------|-------------------------|----------------|------------|------------|--------------|-------|
| E280          | PI                      | PI             | PI         | PI         | 280          | PΙ    |
| E400          | PI                      | PI             | PI         | PI         | 400          | PΙ    |
| Ei280         | LP                      | PI             | PI         | PI         | 280          | PI    |
| Ei400         | LP                      | PI             | PI         | PΙ         | 400          | PI    |

**Response#: Although the ice-sheet extent is prescribed in our experiments, climate feedbacks - such as changes in cloud cover, atmospheric temperature, and moisture transport - alter the surface radiation balance and can reduce albedo in regions like East Antarctica, where the ice-sheet mask remains unchanged. This has been clarified in the results section.**

**3. Improved analysis**

106: the authors state that "when ice-sheet reductions are included there is a persistent positive phase of the SAM, which intensify the westerly winds" On Figure 4, timeseries of SAM indices are shown for all the experiments (note that the x axis is missing), and in all cases the SAM index is much lower in the Ei than E case. So the SAM index is less positive when the ice-sheet reductions are taken into account contrarily to the statement above. I note that even in the PI case (E280), the SAM index is centered on 0.2, which seems odd. The authors should double check their results and include a description of their SAM index calculation.

**Response#: Following this very helpful point, we have rewritten the section in which we detail the**

polar Southern Hemisphere climatic response to the modified boundary conditions. We have recalculated the Southern Annular Mode (SAM) index using EOF analysis of sea level pressure (SLP) over the Southern Hemisphere. The SAM index (PC1) is standardized for comparability across experiments and is now filtered using a Savitzky-Golay filter. We included the following paragraph on the method, as well as new SAM figures (Figures 1 and 2):

"The Southern Annular Mode (SAM) was calculated using EOF analysis to identify the dominant mode of SLP variability in the Southern Hemisphere. This produced the spatial pattern of the first EOF (the SAM pattern). To calculate the time evolution of SAM, a principal component (PC) analysis is used. This (PC1) is obtained by projecting the SLP anomalies onto the EOF pattern. It is then is standardized to have zero mean and unit variance by removing the mean and dividing by the standard deviation. This allows for a direct comparison of SAM variability across different climate experiments."

Figure 1: Southern Annular Mode (SAM) index timeseries for the E280, E400 and Ei400 experiments, for a 200 year equilibrium state period. SAM index is filtered using a Savitzky-Golay filter.

Figure 4 now clearly shows:

- A shift towards a positive SAM phase under increased CO2 atmospheric concentration (E400)
- Amplified SAM variability under combined CO2 and LP ice sheet extent (Ei400) that indicates a dipole
  pattern compatible with a negative SAM.

**#Response#:** We also included anomaly of the zonal wind component to explain the how changes in SAM affect the climate around the polar Southern Hemisphere. The section is now fully rewritten, and we constrain the new SAM analyses with other climatic processes.

114-115: the authors state that there is a "deepening of the Amundsen sea low with stronger winds and stormier conditions", but this is not shown and it sounds to me more like a simple

Figure 2: Difference in EOF first mode of sea level pressure. a) E400-E280, b) Ei400-E280. U component of wind anomalies c)E400-E280, d) Ei400-E280.

description of what is expected under positive SAM than in the simulations presented. If you want to keep that statement, then please show the changes in SLP or geopg in a more convincing way as this is really not evident from Fig. 3 c,d. In addition, the two statement above are contrary to the statement L.163-164: "our simulations do not indicate significant changes in the wind regime over Antarctic and the Southern Ocean." These contradictory statements and figures suggest that the authors need to carefully look at their results and assess whether there are significant changes in the winds or not. Please show the wind changes.

118: The authors suggest that the poleward contraction of the westerlies amplifies the upwelling, which accelerates sea-ice melting from below. Related to the comment above, are there significant changes in the westerlies in your simulations? Please show the wind and upwelling changes.

**Response#: The highlighted statements have been revised for consistency and clarity. We removed the figures describing the SLP anomalies, as our analysis was arbitrary and confusing. In the rewritten section, we now do not make claims without having a careful analysis to support it. By including figures that show changes in the SAM spatial and temporal pattern and in the eastward wind at 850, we confirm substantial differences between E400 and Ei400. Our comments on upwelling are now tied to the compound analysis of zonal wind regime that influences ocean divergence, the spatial patterns of temperature and salinity, and the identification of changes in the clockwise meridional overturning circulation that represents the upwelling of the Circumpolar Deep Water (CDW). We also add strong literature evidence to our discussion.**

Figure 5 does not seem appropriate to discuss AABW changes. I) the AABW in E280 should be shown. Ii) the AABW taken through this calculation seems really low and does not seem to reflect the lower limb of the MOC as seen on Figure 7. Is your value of the global stream function south of 60S and below 500m even negative (ie anticlockwise circulation)? In fact, on your figure 7, the lower limb of the MOC does seem weaker in Ei400 than E400. Coarse resolution models do have issues representing AABW formation, and on Figure 7, there does not seem to be any downwelling branch south of 60S (ie anti-clockwise circulation south of 60S). Please re-assess your statements L. 153-156.

**Response#: We appreciate the comments drawn towards figure 5. Specifically:**

- I. We included a timeseries of the AABW in experiment E280 to highlight the changes in E400 and Ei400 in relation to the control simulation variability (Figure 3).
- II. The values shown, as explained in Figure 5, represent the absolute values of the streamfunction. They are, as expected, negative. However, we chose to represent the circulation in terms of absolute values as the representation in a streamfunction of AABW in negative (anticlockwise), can be misleading in a timeseries. They are also quite low as they reflect the E400 and Ei400 experiments, where the AABW is weaker.
- III. In the statements of given of how the AABW is computed, we mention that it is computed as the absolute value of the minimum at south of 60°S and below 500m depth. Therefore, it does reflect the strength of the AABW in this latitudinal band. The strength seen in the latitudes further north are more related to the export of the AABW to the AMOC, rather than an integration through the polar Southern Hemisphere.
- IV. The discussion has been revised to clarify that Ei400 exhibits a partial recovery of AABW after 700 years, while E400 continues to decline.

170-174: I am confused about this paragraph. Is this a statement based on what has been shown in previous studies or based on the authors' results? If the former, then references are critically needed. If the latter, then this should be shown as it does not seem to be what is obtained here (ie SAM is less positive when the albedo decreases over Antarctica), and references to the figures where it is shown should be included.

Figure 3: Time series of the Antarctic Bottom Water formation index for experiments E280, E400 and Ei400, calculated as the absolute value of the minimum global streamfunction south of 60°S and below 500 m depth (adapted from Zhang et al. (2019)).

**Response#: This paragraph has been revised to clearly distinguish between our results and previous literature. The discussion now refers to the recalculated SAM index and uses references such as Thompson and Wallace (2000) and Gerber and Vallis (2007) to contextualize the SAM's behavior. We have also been careful towards including all references to figures that represent the patterns we depict and hypotheses we propose.**

4. An improved discussion of the results in comparison to previous studies is needed. For example, previous studies looking into the mid-Pliocene (Weiffenbach et al., 2023, Climate Past, https://cp.copernicus.org/articles/20/1067/2024) and Last Interglacial (Yeung et al., 2024, Com. Earth Env., https://www.nature.com/articles/s43247-024-01383-x) highlighted the impact of Southern Ocean warming and reduced sea-ice on stratification and AABW formation, with potential feedbacks on the AIS. The impact of AIS loss on climate was also studied in Hutchinson et al., 2024 (Nat. Com., https://www.nature.com/articles/s41467-024-45501-x). While in that study, changes in orography were also taken into account, it seems relevant to include this study in the discussion as the climatic impact seems much smaller in Hutchinson et al than in your study.

**Response#: Following your suggestions we have included a literature review in the introduction, focusing on paleoclimate insights from the Late Pliocene (LP) as an analogue for future warm climates. This highlights the niche of our study. We have then contrasted our results in the section LP ice sheets as analogues for future climate with other relevant studies. This includes those suggested by yourself for a paleo climate perspective, plus literature examining modern polar Southern Hemisphere climatic change including Goerte et al 2023. We highlight that our larger climate response likely stems from isolating the albedo effect without compensatory feedbacks (e.g., sea-level rise-induced circulation changes). We also highlight the importance of conducting our experiments as to disentangle the contributions of albedo feedbacks from the contributions of change in ice sheet orography that other studies include.**

**Line by line comments:**

60, 61: Please include the references to the mid-holocene, Last interglacial and mid-Pliocene simulations.

**#Response#:** Thank you for reminding us of it. We have now included Zhang et al 2021 (10.5194/gmd-14-1147-2021), Chen et al 2022 (10.1029/2022GL098816), Han et al 2024 (10.1029/2023GL106145) and de Nooijer et al 2020 (10.5194/cp-16-2325-2020) as the relevant references for this section.

124: If I understand correctly, there is no ice-sheet model. Marine ice sheet instabilities parametrization is only relevant for ice-sheet models.

**Response#: We agree this sentence is misleading. We have removed this statement.**

Legend of figure 5: Please include the area or latitudinal band of the Southern Ocean that was used for this graph.

**Response#: We acknowledge that this information is necessary and we have included the latitudinal band of 60-90°S as the region of calculation of AABW index. We have also named this latitudinal band as polar Southern Hemisphere, to properly identify our study region.**

131: the reason for the interannual to decadal variability is not shown here, so this seems like a speculation and it does not seem relevant to the study.

**#Response#:** We agree and have removed this sentence as it was not necessary.

163-164: Please re-assess if this correct after looking into the wind changes (see comment 3 above).

**#Response#:** After analysis of the eastward wind anomalies, as per recommended, we have corrected our statements.

203-208: this paragraph might need to be a bit modified. Even though there are some uncertainties associated with Antarctic topography after the loss of WAIS, it seems that it would be more consistent to do so. I have no problem with the study only looking at the impact of changes in albedo, but the authors have to acknowledge that assessing the impact changes in orography is also important.

**Response#: We have understood your concern and modified the paragraph to:**

"There is a notable gap in existing research on the isolated impact of the abrupt removal of large portions of AIS and GrIS on climate and ocean circulation. The Late Pliocene serves as an important analogue of future climate scenarios due to its modern-like atmospheric  $CO_2$  concentrations, significantly reduced ice sheets and ecosystem shifts. This paleogeographic framework offers a valuable opportunity to assess climate sensitivity to regional albedo changes. In this study, we specifically isolate the albedo effect of ice

sheet reduction to examine its role in driving Antarctic climate dynamics and Southern Ocean circulation. A PlioMIP-conform idealised sensitivity experiment would incorporate the orography changes that would accompany the reduction in ice sheet. Under future ice sheet loss, the changes in orography that will result are likely to have significant impacts on the surrounding climate and Southern Ocean. Here, we choose to not include orographical change however. The additional uncertainties from both deriving the parameters needed to modify orography in the model (including adjusting mean orography in the atmospheric component and sub-grid scale parameters such as standard deviation, slope, and angle) and from whether future orographical changes will closely align with those reconstructed for the Late Pliocene, may outweigh the benefits of an idealised sensitivity experiment. Our simulations with isolated PRISM4D ice sheet conditions demonstrate that ice sheets play a critical role in modulating climate feedbacks in response to warming."

We fully acknowledge, however, that assessing the impact changes in orography is important. We have not yet encountered a sensitivity experiment that looks into this effect in an isolated form. The usual analysis of the Late Pliocene ice sheets in climate simulations includes both altering the ice sheet extent and the orography to match the reconstructed LP conditions. Therefore, our approach introduces novelty by disentangling the effects of albedo from orography and facilitates comparison with the PlioMIP/PMIP ensemble for their Ei experiments.

**Review 2**

In this paper, the authors use the EC-Earth climate model to test the impacts of reduced albedo under Late Pliocene ice sheet configurations to the climate system. Experiments also include the impacts of elevated greenhouse gas levels. While the topic is of great interest, I have several big-picture concerns, specifically about the internal consistency of the experiments presented here.

First, it is unclear what boundary conditions the authors are using for these experiments. In Section 2.2 (Experiment setup), the authors state they are using the PlioMIP 4F Tier 2 experiments (Line 78). PlioMIP uses Pliocene boundary conditions of the ice sheets, including modified ice sheet extent and orography and other changes associated with a reduced ice sheet and warm period, such as changes to vegetation and GHGs etc. However, the previous paragraph (L75) the authors state that they exclude orographic changes and freshwater input associated with the ice sheet extent. If indeed orographic and freshwater changes are not included in these experiments, it remains unclear to me how these changes were implemented in the model. What albedo was chosen in places where West Antarctica or Greenland are longer "ice"? How are these grid cells treated, presumably as land? What albedo is used in these locations? More clarification and explanation is required. Thereafter throughout the manuscript, the authors often refer to using a Pliocene ice sheet configuration, and that such a configuration serves as an analogue of future configurations. The term 'configuration' implies that all relevant elements of that configuration have changed, including orography (e.g. L196). Better explanation and justification is needed on experiment design, and care must be taken when using terms like 'configuration' and 'reduced ice sheet extent' throughout, especially if this is not what the experiment is actually doing.

**#Response#:** We appreciate the questions raised by the reviewer and understand that we needed to provide a more concise and accurate description of our experiment design and motivation for our study. Firstly, we have included in the methodology section, more detailed information of how the model boundary conditions were changed in order to perform our sensitivity experiments:

"The aim of these sensitivity experiments is to unveil the isolated impact of shrinking of the AIS and GrIS to the climate of the polar Southern Hemisphere. Therefore, Ei simulations involve changing only ice sheet extent for the Greenland and Antarctic Ice Sheets (Haywood et al., 2016; Chandan and Peltier, 2018). LP AIS was originally developed using the high-resolution British Antarctic Survey Ice Sheet Model, integrated with climatologies from the Hadley Centre Global Climate Model (Hill et al., 2007; Hill, 2009), utilising PRISM2 boundary conditions (Dowsett et al., 1999). Figure 1 provides a visual comparison of the modern and LP ice sheet extent. LP GrIS reconstruction is provided for PlioMIP2 Haywood et al. (2016) and based on 30 modelling results from the PLISMIP project Dolan et al. (2012).

The PRISM2 ice sheet mask was interpolated onto the grid of EC-Earth's atmospheric component, IFS, and substituted into the snow depth variable of the initial condition file. IFS does not have a specific variable for ice sheets, therefore, altering the snow depth provides only a change in the ice sheet extent, being ice sheets the regions where snow depth exceeds 10 metres. To solely focus on climatic feedbacks resulting from change of AIS and GrIS; albedo values were not modified, modification of ice sheet orography was excluded and accompanying freshwater hosing experiments to account for meltwater from ice sheet change were not performed. With these exclusions, we aim to create idealised sensitivity experiments."

**#Response#:** Following your suggestion, we have replaced the term "configuration" with "extent", as this is the specific variable we modify in our experiments.

**Response#: Regarding the potential for analogy that we have drawn in several sections of our manuscript, we have included sentences that highlight the fact that our findings reveal that the albedo change resulted from reducing AIS and GrIS has an opposite impact as what is found in literature for the LP orography and freshwater input from ice sheet melting. With our experiment design, we were able to disentangle the contribution of this aspect of ice sheet dynamic from other climatic feedbacks. Further research under the umbrella of the PlioMIP project phases has only been performing experiments where the orography and ice sheet extent were modified in parallel. Therefore, with our experiment, we are able to acknowledge the amplified importance of reduced ice sheet height and increased meltwater input to the Southern Ocean in the context of current and future climate change. Our findings reveal that the albedo change as a result of our Ei400 experiment, isolated, acts in the opposite direction as freshwater forcing from ice sheet melting and orography change that are analogous to the Late Pliocene conditions. We derive such conclusions with literature from studies that target the Late Pliocene, current and future climate change. Without performing such an experiment, we would not be able to disentangle these forcings and understand their contributions to climate change.**

**#Response#:** We also acknowledge in our conclusions that future research could aid in directly attributing the role of CO2 forcing, LP ice sheet orography, meltwater forcing and boundary condition modifications onto not only LP but also future climate change:

"This study underscores the critical sensitivity of the Southern Ocean to increased atmospheric warming. Current conditions do not yet reflect the full extent of AIS reduction or collapse, as projected in future climate experiments Roach et al. (2023); Armstrong McKay et al. (2022); Naughten et al. (2023); Steig et al. (2015). However, our results suggest that even under present trends, the ability of the Southern Ocean to ventilate the deep ocean is at significant risk. Furthermore, while we isolate the albedo effect in this study to reduce uncertainties, the exclusion of other climate feedbacks may underestimate the potential catastrophic outcomes of AIS collapse.

This raises potential research questions for future investigation. We believe that an extended set of sensitivity experiments would provide valuable insights into future climate change and even reduce model biases. Experiments would include:"

- 1. Freshwater hosing equivalent to the ice sheet volume that is reduced in LP relative to PI;
- 2. Application of reconstructed LP paleogeography (topography and bathymetry);
- 3. Increased greenhouse gas forcing;
- 4. Interactive ice sheets.

If I understand correctly, only albedo is being modified in these experiments. I am struggling with the rationale and relevance for such an experiment. Keeping a modern AIS/GrIS topography, but reducing albedo to match Pliocene ice extent results in inconsistent boundary conditions. You cannot collapse WAIS as in Figure 1 but maintain modern topography. If WAIS is ice-free this implies it is collapsed. Therefore, not only is the orography fundamentally different, but also ocean pathways between the Ross, Amundsen and Weddell Seas open up, leading to different ocean circulation, sea ice regimes and atmospheric circulation (e.g. Steig et al (2015), Pauling et al (2023)). Therefore, much of the analyses presented here (changes in sea ice regime, SAM index, albedo, surface air temperatures and precipitation) is limited by the presence of land above sea level in West Antarctica, and drawing conclusions about those elements of the climate system would be inaccurate.

**Response#: We acknowledge this limitation and address it directly throughout the paper. We now emphasize that:**

- This is a highly idealized experiment designed to isolate the effect of albedo change;
- Other PlioMIP studies change both orography and extent in parallel, making it hard to separate individual effects;
- Our design provides novel insights by disentangling these often conflicting forcings

**#Response#:** Future work is proposed to explore experiments that combine albedo, orography, and freshwater forcing in a systematic sensitivity framework.

**Response#: We have also clarified in the text manuscript that we do not change the underlying land/sea mask or surface albedo values in the model. Therefore, although the snow depth is reduced below 10 m**

(removing the ice sheet), the model retains the pre-industrial albedo values in those grid cells. This simplification helps us isolate the indirect feedbacks (e.g., cloud, temperature) from the direct albedo effect. This has been explained in detail in the methodology section.

If the goal of the study is to ask how the Southern Ocean and AIS climate responds to reduced ice sheets (as in the manuscript title), this experimental design cannot answer that question. Instead, it seems to me that the authors are asking what the impact would be if WAIS was suddenly painted black (or whatever low ocean/land albedo is being used here). If that is the question, please justify. To isolate the effect of albedo a better tool might be a simple global energy balance model. This would also resolve the authors' technical issues with modifying the atmospheric component of the model when modifying orography (though I will note this is technically possible and not uncommon (see PlioMIP experiments)). Another option would be to use the current experiments, but rather than comparing to a PI control run, also compare this simulation to a run where the full configuration of the ice sheets is changed (including orography). This would allow the authors to better detangle the role of albedo-only changes, by comparing to cases with and without orography change.

**Response#: We appreciate the reviewers suggestions and ideas. We considered an energy balance model as they can help isolate radiative effects, but they lack interactive ocean-atmosphere dynamics. Using an Earth System Model enables us therefore to trace feedbacks affecting winds, SAM, stratification and deepwater formation, which are critical aspects of polar climate. We have strengthened our justification for using EC-Earth3 in this context, and drawn more light to the uniqueness of our study as a fully PlioMIP-conform study with the same model has been published in de Nooijer et al. 2020.**

Once these big picture issues are taken care of, there are a number of other issues throughout the paper that I recommend addressing. For example: As mentioned above, the title does not accurately represent this work – you are not actually reducing ice sheet extents in these experiments

**Response#: The title has now been changed to more accurately represent our main objectives: "Late Pliocene Ice Sheets as an Analogue for Future Climate: A Sensitivity Study of the Polar Southern Hemisphere".**

The abstract needs work to make connections clearer and providing appropriate motivation for the experiment.

**#Response#:** We have re-written the abstract considering your recommendation. The abstract has been revised to better articulate the motivation and clarify that this is an idealised sensitivity experiment focusing on surface reflectivity change, not full PlioMIP-like boundary forcing.

As mentioned above, the experiment design (Section 2) should include a more comprehensive description of the experiments and implementation.

**Response#: We have now included a table with full experiment design (see below) and an in-depth methodology of the changed boundary conditions, as clarified in early comments.**

"To investigate the impacts of varying ice sheet extent and  $CO_2$  concentrations in the polar Southern Hemisphere, we performed a series of sensitivity experiments, displayed in the below table. These experiments employed modern ice-sheet extent (labeled E) and Late Pliocene ice-sheets (labeled Ei) under two atmospheric  $CO_2$  levels: pre-industrial (280 ppmv) and intermediate (400 ppmv)."

Table 2: The four Core and Tier 2 Pliocene for Future protocol experiments conducted. PI refers to preindustrial conditions, LP for Late Pliocene. Name terminology is from Haywood et al., 2016.

| Experiment ID | Ice sheet extent | LSM | Topography | Vegetation | $CO_2$ (ppm) | Orbit |
|---------------|------------------|-----|------------|------------|--------------|-------|
| E280          | PI               | PΙ  | PI         | PI         | 280          | PI    |
| E400          | PI               | PΙ  | PI         | PI         | 400          | PΙ    |
| Ei280         | LP               | PΙ  | PI         | PI         | 280          | PΙ    |
| Ei400         | LP               | PΙ  | PI         | PI         | 400          | PΙ    |

The protocol for our pre-industrial (PI) simulation follows Eyring et al., 2016. Ice sheets, land geography, topography and vegetation are all unmodified from the model. GHG concentrations for  $CO_2$ ,  $CH_4$  and  $N_2O$  are 284.3 ppmv, 808.2 ppbv, and 273.0 ppbv, respectively. For orbital parameters; eccentricity set at 0.016764, obliquity 23.549 and perihelion - 180 is 100.33.

Figures need to be better labeled. For example, Figure 1 should have labels, and the caption should explain more. Consider also including a cross section of the orography for the two ice sheets.

**#Response#:** We have worked carefully in labelling our figures in a clearer manner. We have chosen, however, to not include a cross-section of the orography of the two ice sheets as orography is not modified in this study.

Figure 2 it is unclear why albedo is spatially variable in the Ei experiment. This is confusing.

**Response#: While the prescribed snow depth change is fixed, model feedbacks (e.g., clouds, moisture) alter the effective albedo. This is now clarified in the text accompanying the figure:**

"In Ei400, albedo changes are not confined to regions affected by ice sheet change and there is an overall albedo decline of 20% across the Antarctic interior, with decreasing severity toward the eastern coastline. There is a small hotspot showing pronounced loss of 30-40% on the east coast (60°-70°E, 75°S). Additionally, albedo decreases of up to 30% are observed along the coastline at 0°-10°E and 140°-160°E. These widespread albedo reductions are a result of the interplay of climate feedbacks, with changes in cloud cover, atmospheric temperature and moisture transport influencing the radiation balance and surface reflectivity."

There are some claims throughout, such as the stability of ice sheets being increasingly at risk, please provide appropriate citations for these.

**Response#: Appropriate references have been added to support such statements, including recent work on AIS vulnerability and Pliocene analogues.**

Usually Greenland Ice Sheet is abbreviated as GrIS rather than GIS.

**#Response#:** This has been corrected throughout.

Include references when discussing EC-Earth's proven capabilities.

**Response#: We have now included, per your recommendation, the references: Hang et al 2021, Chen et al 2022, Han et al 2024 and de Nooijer et al 2020, to prove EC-Earths capabilities over different paleo time periods.**

The discussion requires more work, more context, and some of the concluding claims are not well-supported. For example, it is not clear how this experiment will help identify early signals of ice sheet retreat and possibly ice sheet instabilities because this study does not include a dynamical ice sheet.

**Response#: Following the new analysis, improved understanding of atmospheric and ocean processes and properly highlighting the uniqueness of our study, the discussion section has now been rewritten. It now links together our results, the interplay between them and compares the results and mechanisms with other studies, including paleo Antarctic work (Weiffenbach et al., 2024, Hutchinson et al., 2024, Yeung et al., 2024), and literature focusing on current and future polar southern hemisphere change (Silvano et al., 2023, Sidorenko et al., 2021, Kidston et al., 2011, Kusahara et al., 2017). We have fully delved into the limitations of the experimental setup and avoid unsupported conclusions. We now clarify that this experiment isolates the albedo effect, and future studies should include meltwater and orography to understand ice sheet instabilities. This is reflected in the proposed set of follow-up experiments.**

---

## Referee Report (RR1)

**Late Pliocene Ice Sheets as an Analogue for Future Climate: A Sensitivity Study of the Polar Southern Hemisphere**

The authors set out to assess the impact perturbing Antarctic and Greenland albedo has on the polar Southern Hemisphere climate. The paper includes no analysis of any  $GrIS_{ANOM}$  impacts; this should be reflected in both the introduction and results sections.

The description of the experimental design is lacking key details.

- 1. Considering Table 1; are all simulations initialised using the same spun-up pre industrial (PI) ocean state? Thus, E400 is an instantaneous N×PI increase in CO2? This would mean that Ei280 is also not spun up (a simulation which doesn't seem to be analysed anyway; does it need to be included?).
- 2. The Pliocene Antarctic and Greenland ice extents are mapped onto PI model states via adjustments land surface albedo. It is not clear how, by reducing snow depth, the albedo is modified. As I understand it, EC-Earth3 considers snow over certain depth to be perennial; with a fixed albedo and no accumulation allowed (a proxy for ice sheet surface conditions). If the snow depth is reduced below this threshold, is the surface considered to be seasonal, with albedo a weighted function of  $\alpha_{snow}$  and  $\alpha_{rock}$ ? With WAIS orographic height retained, I would expect multi-year snow to accumulate (unless this is inhibited, as per the perennial snow); maybe that is what is happening (see Fig.2(d), WAIS  $d\alpha = 0$  regions).
- 3. The authors need to justify the usefulness of effectively transforming ice into rock (change of albedo, but no change of orography). What physical information does this provide (beyond what could be obtained from a simple energy-balance model)?

**Results**

- 1. The comparison of E400 with E280 is essentially a comparison of contemporary climate with PI (albeit, with an instantaneous CO2 increase). The resulting Antarctic temperature changes are large when compared to observations or other modelling studies; this seems to be a result of excessive sea-ice loss. Is this similar to that seen previous EC-Earth3 historical simulations? The changes observed in Ei400 are harder to interpret without fully understanding the albedo adjustment (see above). The inland signals are dominated by the warming over the Ross, Ronne and Amery ice shelves; this is understandable as these are low altitude (warm) regimes that have been transformed into storage heaters.
- 2. Differencing of same field regressed principal component loading patterns (Fig 3(a,b)) is unusual, and difficult to interpret without any measure of significance. Does EOF1 in the three regimes represent a similar percentage of variance? If not, these differences would be problematic. Rather than (or in addition to) the EOF formulation, SAM indices of the  $P_{lat1} P_{lat2}$  form would be easier to compare. Figure 3(c) shows an apparent intensification of SAM, but without the normal peninsular warming; the following ocean analysis also shows a decrease in on shelf CDW (the opposite to that expected). It is suggested that E400i has a more chaotic pattern; this isn't reflected in Figure 4(c) (again variance measures are required). The wind pattern (Fig. 3(d)) is indicative of a weaker SAM, with a shift of the jet to more northern latitudes.
- 3. The AABW recovery in Ei400 is of interest, and the mechanisms should be analysed further. There is an increase in salinity at depth, but this is associated with an increase in temperature; it would be beneficial to see the density profiles.

**General:**

- 1. When showing difference plots, show some measure of significance.
- 2. Line #199; missing figure number.

---

## Author Response (AR2)

**Author response for egusphere-2024-4061**

Katherine Power, Fernanda Matos, Qiong Zhang

August 19, 2025

Dear Editors and Reviewers,

We are extremely pleased that the editor and reviewers are satisfied following the first round of major revisions. With this round of revisions, we have aimed to significantly improve the clarity of the experiment design. This includes a more extensive methodology to explicitly clarify what and how we are testing, plus the removal of ambiguous wording, with the aim of the study clearly and concisely reiterated throughout. We have taken onboard the comments regarding the EOFs, and performed new, clearer analysis, taking into consideration the variance of the different experiments. This is accompanied with thorough statistics, helping to guide the reader further. We now delve deeper into deep water changes, with new analysis including; Hovmöller diagrams of salinity and temperature anomalies at  $60^{\circ}$ S and Potential temperature ( $\theta$ )—salinity (T–S) diagrams of the Southern Ocean. We have also restructured and improved the discussion and conclusion, providing a clearer framing to better understand the mechanisms involved and contextualise our results relative to previous studies. We hope, with this new revision of the manuscript, we have addressed all remaining concerns, deeming the paper publication worthy.

Below follows a reply to each reviewer comment.

Sincerely,

Authors

**Reviewer 1**

Abstract: Changes to GrIS are prominent in the abstract. I understand that the GrIS albedo was also changed, however it is not shown in the manuscript, and quite unclear whether GrIS changes contribute to the pSH climatic changes discussed in the manuscript. I am not saying that you should not mention GrIS changes, however if you do not think that GrIS changes impact the pSH, then you might tone down the GrIS in the abstract.

**Response#: Thank you for highlighting, we have now changed the tone of the abstract to avoid focusing too much on the GrIS. It now reads:**

"The Earth's ice sheets, including the Antarctic Ice Sheet (AIS), are critical tipping points in the climate system. In recent years, the potential future collapse has garnered increased attention due to its cascading effects, that could significantly alter global climate patterns, and cause large-scale, long-lasting, and potentially irreversible changes within human timescales. This study investigates the large-scale response of the polar Southern Hemisphere (pSH; comprising the Southern Ocean and Antarctica (60-90°S)) to the geometric reduction of ice-sheets to a reconstructed Late Pliocene (LP) extent, and imposing increased greenhouse gas forcing in the Earth System. Using the PRISM4D reconstruction, where ice sheets such as the West Antarctic Ice Sheet (WAIS) were significantly diminished, we conducted multi-centennial simulations with the EC-Earth3 model at atmospheric CO2 concentrations of 280 ppmv and 400 ppmv."

**L. 30: "oceanic" circulation**

**#Response#:** Altered

L. 33: Is there really reduction of snowfall on the AIS and GrIS atm? There is more snowfall on EAIS. Please make sure this statement is precise.

**Response#: Thank you for pointing out the broad tone, the sentence is now clearer:**

"However, the stability of these ice sheets is currently at risk due to climate change driven enhanced surface and basal melting, with melting rates projected to intensify in the following decades".

L. 43: MIS5e is mentioned here at 125,000 yrs ago, while in the discussion you mention the LIG. Please pick one or the other and use it consistently throughout the manuscript, noting that the PMIP LIG timeslice is at 127ka.

**Response#: To avoid inconsistencies we now only refer to the LIG.**

**Figure 1: What do the green lines represent? Please include it in the caption.**

**Response#: Thank you for bringing this to our attention, the greenlines were used previously as an outline but looked messy and were not necessary to include here. We have reproduced a clearer figure (figure ??) shows the ice mask, plus the continental outline of modern Antarctica to better understand the changes in extent of the ice mask.**

**Figure 4: Some statistics on the SAM indices are needed as it is hard to compare the 3 simulations in this way.**

**Response#: We appreciate the comment and agree that the PC1 raw indexes were tricky to decipher. After reexamining the previous EOF analysis, we realised a crucial step had been missed in the processing of the data. Therefore we have reconducted EOF analysis, including a regression analysis of the seasonal zonal wind anomalies onto the first principal component (PC1) representing the SAM index. The new results are displayed in Figure 3, complete with variance. Reperforming the EOF calculation gave insight to much clearer patterns and results, which we now discuss with detail. With regards to your comment, we have also included a comprehensive statistics table showing variability, kurtosis and occurrence percentages of positive/negative and extreme events, describing the temporal behavior of the SAM PC1 in each experiment (Table 1). The new PC1 figure (figure 4) now also includes a 10 year smoothed running mean to highlight**

multi-decadal variability. We have, therefore, re-written this section.

|       | Median | Std Dev   | Kurtosis | Positive (> | Negative (< | Extreme Posi-      | Extreme Neg-        |
|-------|--------|-----------|----------|-------------|-------------|--------------------|---------------------|
|       |        |           |          | 0)          | 0)          | tive $(>+1\sigma)$ | ative $(

Figure 3: Hovmöller diagrams of salinity (a, b) and temperature (b, d) anomalies at  $60^{\circ}$ S, relative to E280 average at the same latitude, for E400 (a, b) and Ei400 (c, d).

**Response#: In addition, the T–S diagrams (Figure 4) show that AABW, represented by the deepest water masses (red), forms under warmer, saltier, and lighter conditions in E400 and Ei400 compared to E280, which is expected due to the imposed greenhouse gas forcing. However, relative to E400, the AABW in Ei400 is characterized by higher salinity and a broader temperature range, consistent with its gradual recovery. Finally, the vertical anomaly structures in salinity and temperature and the expansion of AABW formation shown in these figures provide a coherent recovery mechanism: enhanced surface freshening in Ei400 strengthens the halocline, isolating the ocean interior, which subsequently becomes saltier. The deep ocean warming is not as strong as to counteract this salinization effect in Ei400, since it happens progressively after an initial cooling of the ocean interior and limited to about 1°C in the deep ocean. In E400, on the other hand, this warming exceeds 1°C, the isolation effect of the upper ocean is not as strong, and the salinization of the deep ocean is not enough to enable strong AABW formation.**

Figure 4: Temperature–salinity (T–S) diagrams of the Southern Ocean for E280 (a), E400 (b), and Ei400 (c), averaged over the last 200 simulation years

**Response#: Regarding the question about the downwelling of the AABW branch south of 65°S not being displayed as a counter clockwise circulation, it occurs simply because of the basin-scale integration that is done to derive the streamfunction output we use to display the overturning. In our simulations, the near-Antarctic overturning south of 65°S is weaker than 1.5 Sv in magnitude, thereby being partially cancelled in the basin-integrated streamfunction. An alternative would be to derive the overturning at different basins, which is out of the scope of our study. Nevertheless, the exported counterclockwise AABW cell is evident north of 65°S below 3000 m. Thus, the apparent red shading south of 65°S reflects the basin integration, not the absence of AABW formation.**

L. 385-388: I have no problem with your experimental design, ie just changing the albedo, however I do not agree with stating that completely adjusting the AIS (and GrIS) to LP would lead to additional uncertainties, more over when you conclude (L 420) that the LP paleogeography should be implemented.

**Response#: Thank you for this point. We have revised the manuscript to clarify that this methodology is a deliberate, scope-limited choice rather than claim that orographic modifications are unimportant. We now present application of reconstructed Late Pliocene paleogeography as a complementary follow-up experiment rather than as an argument against it. In the restructuring of the discussion and conclusion sections, this is now within the conclusion and therefore has a improved flow to future work. Additionally, following the recommendations of reviewer 2, we include an example of a possible impact of orography change, L408-426.**

L. 404-405: Please be more precise with respect to the results of Gorte et al. Which time period are they looking at? Why do they simulate weaker AABW? ... In general, a more precise text with clearer conclusions with respect to the mechanisms would be beneficial for the paragraph 397-406. ie you could more precisely summarise what previous studies concluded with respect to the impact of i) meltwater input in the SO, ii) changes in AIS topography, iii) changes in SO stratification due to SO warming. You can then compare the results from iii) with your results and implications from currently missed processes i and ii in your simulations. #Response#: Following this very helpful suggestion we have restructured the discussion section, L320-

400. This clearer framing improves the mechanistic interpretation and contextualises our results relative to previous studies. We discuss the overall occurrence and implications of changes in ocean circulation regimes during past warm periods, before delving into individual mechanisms with appropriate literature and geological reference. We then discuss the findings of our study and how these compare to previous investigations. The discussion specific to AABW, including the structure recommendations is found L368-400.

**L. 420-424: Instead of bullet points, these could be better linked to the text L. 397-406.**

**Response#: Thank you for the helpful suggestion regarding the presentation of future research directions. We have revised the manuscript to integrate the list of proposed sensitivity experiments into the narrative text for better flow and clarity, rather than presenting them as bullet points, L420-426.**

**Reviewer 2**

It is still not clear to me what is being modified between the PI and LP experiments and what albedo effect you are testing. In L122 it says the albedo values are not modified and in your response you say 'the model retains the pre-industrial albedo values in those grid cells', referring to the grid cells that are no longer considered 'ice sheet'. Does this mean you are not testing the impact of reducing the surface albedo that comes from reducing ice sheet extent (direct albedo) but instead are only testing the indirect albedo changes due to LW energy balance changes? If this is the case, why do you say in the abstract L20 that you isolate the 'direct albedo effect'. I think it need to be explained more clearly what the effect is you are investigating, what is changed between the grid cells that were considered ice at the PI but not at the LP, and include justification for only testing the indirect effect when direct ice-albedo feedbacks are shown to be very important in modifying climate.

**Response#: Thank you for highlighting the need for a more detailed explanation here. To clarify, our sensitivity experiments isolate the climatic response to the geometric reduction of ice sheet extent while holding surface albedo and orography fixed at pre-industrial values. Therefore, we neither prescribe Late Pliocene-specific albedo changes or vegetation feedbacks, nor do we include freshwater forcing or topographic changes. This approach allows us to focus on the radiative and dynamical impacts of ice sheet retreat under a fixed albedo framework, avoiding the additional uncertainties inherent in reconstructing paleogeographic surface properties. We have updated the manuscript accordingly to reflect this distinction. To avoid ambiguity we no longer refer to the "albedo effect" and all uses of it, including in the abstract have been updated to appropriate terminology. We have revised and extended the methods to explicitly state what we have changed and clearly describe what we are testing L141-163:**

L7 and L380 say that LP 'paleogeography' is used which is not completely accurate as orography/bathymetry isn't used only the ice extent.

**#Response#:** Thank you for pointing this out, we agree the term is misleading. It is now only used when discussing future possible experiments, and not in relation to the experiments of this study.

**L131: why do you run the simulations for 1450 years, is it to equilibrium?**

**Response#: Our sensitivity experiments, meaning the E280, E400, and Ei400 were all started in parallel after branching off from a quasi-equilibrated PI spinup spanning 800-years. We defined quasi-equilibrium as a global surface air temperature trend of less than 0.05 K per century. The extended integration allows us to examine multi-centennial variability, with a good compromise between the level of information and output frequency required to our simulations, and computational costs therein. However, as we analyze deep water formation regimes, we do not expect full equilibration of the ocean in the E400 and Ei400 experiments, since large-scale ocean circulation exhibit variability ranging from synoptic to multi-centennial and is subject to self-modulation. We have now clarified the spin up period in the manuscript, L112-118:**

All simulations were started in parallel after branching off from a quasi-equilibrated PI spinup spanning 800-years, to ensure consistent baseline conditions and that any changes observed in the simulations are due to the perturbation and not model drift. We defined quasi-equilibrium as a global surface air temperature trend of less than 0.05 K per century. Thus, the E280 experiment represents our Pre-Industrial control simulation, which is a core experiment of PlioMIP2/3, while E400 and Ei400 represent our sensitivity experiments, being comprised within the Tier 2 experimental design of PlioMIP2 and continuing as optional but pivotal experiments in PlioMIP3 (Haywood et al., 2020, 2024).

L164: Figure ?? needs correcting #Response#: Corrected with Figure 2

L275: Figure ?? needs correcting #Response#: Corrected with Figure 7

L377-380: I think this discussion on gaps in existing research and how and why the LP is a good analogue for future climate scenarios could also be touched on more in the introduction to provide more context (e.g. what the GHG and temp levels were, evidence for how similar the ice sheet extents might be).

**Response#: We agree that adding this context would strengthen the introduction, and have now included comprehensive additional background on the LP, including specifics for Southern Hemisphere. This provides a clearer rationale for using the LP as an analogue for future climate change. We have revised the introduction to explicitly outline the limited number of studies focusing on Southern Ocean and Antarctic mechanisms during the LP, and state how our study addresses this knowledge gap by isolating the albedo effect of Antarctic ice-sheet reduction on the climate of the polar Southern Hemisphere, L40-72.**

L384: Give an example of what the impact of including orography changes might be, include reference.

**Response#: We have revised the text to include an example of the potential impacts of orography changes, L410-413.**

L428: how do you finding help identify early signals of ice sheet retreat? This isn't touched on or discussed anywhere else.

**#Response#:** We agree that the link to early signals of ice-sheet retreat was not sufficiently explained and have removed it from the text:

"These insights are critical for refining future climate models, offering a clearer picture of the potential pathways and risks associated with polar ice sheet instability"

**Reviewer 3**

The paper includes no analysis of any GrISANOM impacts; this should be reflected in both the introduction and results sections.

**#Response#:** Thank you for highlighting this discrepancy. We have removed most of the text that mentions the GrIS to avoid misleading the readers, as the focus of the paper is in the pSH, and other work already exists discussing the ramifications of changing the GrIS. This we mention in the introduction.

Considering Table 1; are all simulations initialised using the same spun-up pre industrial (PI) ocean state? Thus, E400 is an instantaneous  $N \times PI$  increase in  $CO_2$ ? This would mean that Ei280 is also not spun up (a simulation which doesn't seem to be analysed anyway; does it need to be included?).

**Response#: All simulations are initialized directly from quasi-equilibrated PI spinup simulation spanning 800-years to ensure consistent baseline conditions and that any changes observed in the simulations are due to the perturbation (e.g., CO2 increase). This has been clarified in the text L12-115. The E400 experiment represents an instantaneous increase of CO2 concentration from PI (280 ppmv) to 400 ppmv, applied directly during branching off from the spin up. This approach isolates the CO2 effect on the climate system without allowing the ocean and atmosphere to adjust gradually beforehand. This is discussed with detail L116-120. We have removed our mentioning of the simulation Ei280, as it has not been analysed in our manuscript.**

The Pliocene Antarctic and Greenland ice extents are mapped onto PI model states via adjustments land surface albedo. It is not clear how, by reducing snow depth, the albedo is modified. As I understand it, EC-Earth3 considers snow over certain depth to be perennial; with a fixed albedo and no accumulation allowed (a proxy for ice sheet surface conditions). If the snow depth is reduced below this threshold, is the surface considered to be seasonal, with albedo a weighted function of  $\alpha$ snow and  $\alpha$ rock? With WAIS orographic height retained, I would expect multi-year snow to accumulate (unless this is inhibited, as per the perennial snow); maybe that is what is happening (see Fig.2(d), WAIS  $d\alpha = 0$  regions).

**Response#: We thank the reviewer for this detailed question about how snow depth reductions affect albedo in EC-Earth3 when mapping LP ice extents onto the PI model. The snow scheme in EC-Earth3 (based on ECMWF's HTESSEL land surface model) treats snow as perennial when snow depth exceeds a threshold (0.5 m of water equivalent), assigning a fixed high albedo and preventing further accumulation, effectively acting as a proxy for ice sheets. When snow depth is below this threshold, snow is considered seasonal and surface albedo is calculated as a weighted average between snow albedo and underlying surface albedo, reflecting seasonal snow cover variability (Döscher et al., 2022; Balsamo et al., 2009). In our experiments, the West Antarctic Ice Sheet (WAIS) orography is retained, but in regions where snow depth falls below**

the perennial threshold, seasonal snow processes dominate, allowing accumulation and melt with seasonally varying albedo. This leads to heterogeneous surface albedo changes, including areas with little or no albedo change, as visible in Fig. 2(d). Thus, reducing snow depth modulates surface albedo by shifting grid cells between perennial snow (high fixed albedo) and seasonal snow (variable albedo), providing a physically consistent approach to represent ice retreat and its radiative effects in the model. This has now been included in the methods section for further clarity, L141-152.

The authors need to justify the usefulness of effectively transforming ice into rock (change of albedo, but no change of orography). What physical information does this provide (beyond what could be obtained from a simple energy-balance model)?

**Response#: We appreciate the reviewer's question about the physical justification for changing albedo without changing orography. This experimental design is intentional and serves to isolate the climatic response to the geometric removal of ice cover while keeping topography and other factors fixed. Specifically, by retaining pre-industrial (PI) albedo and topography values for grid cells where ice is removed, the model uses its default surface properties for that grid cell type, whether bare land or ocean, rather than applying more complex LP specific albedo or vegetation reconstructions. This approach allows us to focus on the first-order radiative effect. By doing so, we isolate both the radiative effect of ice loss, i.e., how the change in surface reflectivity influences the local and regional energy balance, and the dynamical response, including how these changes affect atmospheric circulation, wind patterns, and ocean feedbacks within the model framework. This controlled experimental design avoids confounding influences from additional forcings such as changes in vegetation, soil moisture, or orography, which may otherwise obscure the direct climatic impact of ice retreat. In this way, the experiment acts as a valuable idealized sensitivity test that serves as a baseline for understanding the isolated role of ice sheet retreat on polar Southern Hemisphere climate dynamics—insights that cannot be easily obtained from simplified energy-balance models that lack interactive atmospheric and ocean dynamics. We have ensured to specify this in the methodology section L153-163, ensuring our experiment design and purpose is clear.**

The comparison of E400 with E280 is essentially a comparison of contemporary climate with PI (albeit, with an instantaneous  $CO_2$  increase). The resulting Antarctic temperature changes are large when compared to observations or other modelling studies; this seems to be a result of excessive sea-ice loss. Is this similar to that seen previous EC-Earth3 historical simulations? The changes observed in Ei400 are harder to interpret without fully understanding the albedo adjustment (see above). The inland signals are dominated by the warming over the Ross, Ronne and Amery ice shelves; this is understandable as these are low altitude (warm) regimes that have been transformed into storage heaters.

**Response#: The E280 and E400 experiments correspond to pre-industrial (PI, 280 ppmv CO2) and contemporary (400 ppmv CO2) atmospheric CO2 concentrations, respectively. The E400 experiment involves an instantaneous jump in CO2 concentration to 400 ppmv and represents a "PI with elevated CO2" scenario. This setup is part of the "Pliocene for Future" (P4F) agenda, designed to isolate and understand the effect of CO2 changes alone on the climate system. Specifically, E400 was a Tier 2 experiment in PlioMIP2 and continues as an optional, but of high-importance, experiment in PlioMIP3 (Haywood et al., 2020, 2024), designed for forcing factorization analyses. Therefore, its primary purpose is to clarify how an elevated CO2 level, without other boundary condition changes, affects climate, helping to separate CO2 driven changes**

from other paleoclimate forcings. We have improved the experiment description accordingly (L116-120).

**Response#: We acknowledge that the Antarctic temperature increases observed through our findings are larger than those reported in observational studies and other modeling efforts. This is consistent with known biases in the EC-Earth3 model where enhanced warming is closely associated with significant sea-ice loss, amplifying regional warming. Döscher et al. (2022) demonstrates this, showing the EC-Earth3 historical ensemble underestimates maximum and minimum Antarctic sea-ice extent by approximately 5 and 2 million km² respectively, primarily due to a warm bias in the Southern Ocean. This underestimation leads to reduced sea-ice cover, which in turn contributes to higher surface temperatures and further sea-ice loss. This has been discussed in the discussion section. However, we would like to emphasize that E400 does not configure a modern climate experiment as per CMIP6 guidelines, since this is represented by the historical simulation, which includes many other radiative forcings through a transient approach. Therefore, comparisons from our results in this manuscript can only be completely made with reconstructions, respecting the evidence of biases also included in them, which does not affect the quality of our results and the significance of our modelling effort to the community. We discuss this L337-345.**

**Response#: We also acknowledge that the climatic changes observed in the Ei400 experiment are more complex to interpret due to the albedo adjustment applied. As explained above, the adjustment in Ei400 involves changing surface reflectivity by effectively removing ice cover without altering orography or vegetation, isolating the radiative and dynamical impacts of ice loss alongside CO2 increase. While this approach clarifies the direct effect of ice retreat on regional climate, it also might introduce nonlinear feedbacks and interactions within the atmosphere and ocean that can complicate interpretation. Ei400 should be viewed as an idealized sensitivity test rather than a fully realistic paleoclimate scenario, with the goal of disentangling individual processes. Finally, the observed inland Antarctic warming pattern dominated by the Ross, Ronne, and Amery ice shelves is consistent with physical expectations, as these low-altitude, relatively warm ice shelves act as heat reservoirs, facilitating heat transfer inland. We have highlighted this interpretation in our revised manuscript accordingly, L336-338.**

Differencing of same field regressed principal component loading patterns (Fig 3(a,b)) is unusual, and difficult to interpret without any measure of significance. Does EOF1 in the three regimes represent a similar percentage of variance? If not, these differences would be problematic. Rather than (or in addition to) the EOF formulation, SAM indices of the Plat1 - Plat2 form would be easier to compare. Figure 3(c) shows an apparent intensification of SAM, but without the normal peninsular warming; the following ocean analysis also shows a decrease in on shelf CDW (the opposite to that expected). It is suggested that E400i has a more chaotic pattern; this isn't reflected in Figure 4(c) (again variance measures are required). The wind pattern (Fig. 3(d)) is indicative of a weaker SAM, with a shift of the jet to more northern latitudes.

**Response#: Thank you for your insightful comments regarding the interpretation and comparison of the SAM patterns and variability across experiments. We agree that direct differencing of EOF loading patterns without accompanying statistical metrics can be difficult to interpret. After reexamining the previous EOF analysis, we realised a crucial step had been missed in the processing of the data, resulting in our previous**

results. We have reconducted EOF analysis, and now include a regression analysis of the seasonal zonal wind anomalies onto the first principal component (PC1) representing the SAM index. This provides a much improved insight into atmospheric changes. The new results are displayed in Figure 3, complete with variance. We have also included a comprehensive statistics table, describing the temporal behavior of the SAM PC1 in each experiment Table 1. Re-performing the EOF calculation gave insight to much clearer patterns and results, which we now discuss with detail. This clears up the opposite to expected effect surrounding the CDW, as now with increased CO2 we establish a shift to a negative SAM phase - associated with reduced upwelling of cold, deep ocean water onto the Antarctic continental shelf. Furthermore, the wind and EOF patterns now align (Figure 5) and facilitates the interpretation of the transfer of near-surface climate response in our idealized experiments to the deep ocean.

Figure 5: The Southern Annular Mode mean state as the first EOF of the SLP in hPa, for the a) PI Control, b) E400 and c) Ei400 experiments with % of variance explained by EOF1 notated for each. Regression of the austral summer (DJF) 850 hPa zonal wind onto the leading SAM principal component for d) E280, e) E400 and f) Ei400. Colors indicate the regression coefficient (m s-1 per SAM unit). Black contours denote statistically significant values (p < 0.05). % of variance explained by EOF1

The AABW recovery in Ei400 is of interest, and the mechanisms should be analysed further. There is an increase in salinity at depth, but this is associated with an increase in temperature; it would be beneficial to see the density profiles.

**Response#: We thank the reviewer for this constructive suggestion. We note that this point is closely**

related to Reviewer 1's comment regarding the interpretation of the recovery mechanism. In response, we have expanded our analysis and included additional diagnostics (Hovmöller diagrams (Figure 3) and T–S diagrams (Figure 4) that directly address the combined salinity and temperature changes and their implications for density. These are discussed L284-318. These figures demonstrate that AABW in Ei400 forms under warmer and saltier conditions than in E280, but that compared to E400 it is saltier and spans a broader temperature interval, consistent with a gradual recovery. We therefore believe that the new analysis requested by Reviewer 1 also covers the reviewer's helpful suggestion here.

**When showing difference plots, show some measure of significance**

**Response#: Significance tests have been performed for all difference plots, with only results statistically significant at the 95% confidence level displayed. This is now added into the figure captions.**

**Line 199; missing figure number**

**#Response#:** Corrected

**References**

Balsamo, G., Beljaars, A., Scipal, K., Viterbo, P., Van Den Hurk, B., Hirschi, M., and Betts, A. K. (2009). A Revised Hydrology for the ECMWF Model: Verification from Field Site to Terrestrial Water Storage and Impact in the Integrated Forecast System. *Journal of Hydrometeorology*, 10(3):623–643.

Döscher, R., Acosta, M., Alessandri, A., Anthoni, P., Arsouze, T., Bergman, T., Bernardello, R., Boussetta, S., Caron, L.-P., Carver, G., Castrillo, M., Catalano, F., Cvijanovic, I., Davini, P., Dekker, E., Doblas-Reyes, F. J., Docquier, D., Echevarria, P., Fladrich, U., Fuentes-Franco, R., Gröger, M., v. Hardenberg, J., Hieronymus, J., Karami, M. P., Keskinen, J.-P., Koenigk, T., Makkonen, R., Massonnet, F., Ménégoz, M., Miller, P. A., Moreno-Chamarro, E., Nieradzik, L., van Noije, T., Nolan, P., O'Donnell, D., Ollinaho, P., van den Oord, G., Ortega, P., Prims, O. T., Ramos, A., Reerink, T., Rousset, C., Ruprich-Robert, Y., Le Sager, P., Schmith, T., Schrödner, R., Serva, F., Sicardi, V., Sloth Madsen, M., Smith, B., Tian, T., Tourigny, E., Uotila, P., Vancoppenolle, M., Wang, S., Wårlind, D., Willén, U., Wyser, K., Yang, S., Yepes-Arbós, X., and Zhang, Q. (2022). The EC-Earth3 Earth system model for the Coupled Model Intercomparison Project 6. Geoscientific Model Development, 15(7):2973–3020.

Haywood, Tindall, J. C., Dowsett, H. J., Dolan, A. M., Foley, K. M., Hunter, S. J., Hill, D. J., Chan, W.-L., Abe-Ouchi, A., Stepanek, C., Lohmann, G., Chandan, D., Peltier, W. R., Tan, N., Contoux, C., Ramstein, G., Li, X., Zhang, Z., Guo, C., Nisancioglu, K. H., Zhang, Q., Li, Q., Kamae, Y., Chandler, M. A., Sohl, L. E., Otto-Bliesner, B. L., Feng, R., Brady, E. C., von der Heydt, A. S., Baatsen, M. L. J., and Lunt, D. J. (2020). The Pliocene Model Intercomparison Project Phase 2: large-scale climate features and climate sensitivity. Climate of the Past, 16(6):2095–2123.

Haywood, A. M., Tindall, J. C., Burton, L. E., Chandler, M. A., Dolan, A. M., Dowsett, H. J., Feng, R., Fletcher, T. L., Foley, K. M., Hill, D. J., Hunter, S. J., Otto-Bliesner, B. L., Lunt, D. J., Robinson, M. M., and Salzmann, U. (2024). Pliocene Model Intercomparison Project Phase 3 (PlioMIP3) – Science plan and experimental design. Global and Planetary Change, 232:104316.